# Pain, Complex Chronic Conditions and Potential Inappropriate Medication in People with Dementia. Lessons Learnt for Pain Treatment Plans Utilizing Data from the Veteran Health Administration

**DOI:** 10.3390/brainsci11010086

**Published:** 2021-01-11

**Authors:** Bettina S. Husebo, Robert D. Kerns, Ling Han, Melissa Skanderson, Danijela Gnjidic, Heather G. Allore

**Affiliations:** 1Centre for Elderly and Nursing Home Medicine, Department of Global Public Health and Primary Care, University of Bergen, 5020 Bergen, Norway; 2Municipality of Bergen, 5020 Bergen, Norway; 3Departments of Psychiatry, Neurology and Psychology, Yale University, New Haven, CT 06511, USA; robert.kerns@yale.edu; 4Pain Research, Informatics, Multimorbidities and Education (PRIME) Center, VA Connecticut Healthcare System, West Haven, CT 06516, USA; Melissa.Skanderson@va.gov; 5Section of Geriatrics, Department of Internal Medicine, School of Medicine, Yale University, New Haven, CT 06511, USA; ling.han@yale.edu (L.H.); heather.allore@yale.edu (H.G.A.); 6Charles Perkins Centre, Faculty of Medicine and Health, School of Pharmacy, University of Sydney, Sydney 2006 NSW, Australia; danijela.gnjidic@sydney.edu.au; 7Department of Biostatistics, School of Public Health, Yale University, New Haven, CT 06511, USA

**Keywords:** pain, dementia, veterans, complex chronic conditions, potential inappropriate medications

## Abstract

Alzheimer’s disease and related dementias (ADRD), pain and chronic complex conditions (CCC) often co-occur leading to polypharmacy and with potential inappropriate medications (PIMs) use, are important risk factors for adverse drug reactions and hospitalizations in older adults. Many US veterans are at high risk for persistent pain due to age, injury or medical illness. Concerns about inadequate treatment of pain—accompanied by evidence about the analgesic efficacy of opioids—has led to an increase in the use of opioid medications to treat chronic pain in the Veterans Health Administration (VHA) and other healthcare systems. This study aims to investigate the relationship between receipt of pain medications and centrally (CNS) acting PIMs among veterans diagnosed with dementia, pain intensity, and CCC 90-days prior to hospitalization. The final analytic sample included 96,224 (81.7%) eligible older veterans from outpatient visits between October 2012–30 September 2013. We hypothesized that veterans with ADRD, and severe pain intensity may receive inappropriate pain management and CNS-acting PIMs. Seventy percent of the veterans, and especially people with ADRD, reported severe pain intensity. One in three veterans with ADRD and severe pain intensity have an increased likelihood for CNS-acting PIMs, and/or opioids. Regular assessment and re-assessment of pain among older persons with CCC, patient-centered tapering or discontinuation of opioids, alternatives to CNS-acting PIMs, and use of non-pharmacological approaches should be considered.

## 1. Introduction

Dementia is a progressive, irreversible brain syndrome that gradually affects the memory, thinking and daily functioning of those who suffer from it [1]. Currently, more than 47 million people worldwide live with dementia. This number is projected to triple by 2050, especially in low- and middle-income countries, where it is projected that 75% of all cases will be located [1].

Emotional disturbance, speech disruption, and acute or persistent pain are common. During the course of dementia, up to 90% develop behavioral and psychological symptoms (BPSDs) including agitation, depression and sleep disturbances [2]. Most people with dementia also have multimorbidity, which makes dementia a complex chronic condition (CCC) that requires individually-tailored assessment and treatment [3]. Altogether, dementia, pain and CCC increase the risk for polypharmacy and subsequently for potential inappropriate medication (PIM) use, an important risk factor for adverse drug reactions and hospitalizations (Figure 1) [4].

Undiagnosed and untreated acute and persistent pain is common in home-dwelling people with dementia [5]. Pain is often related to musculoskeletal diseases, CCC, infections or injuries, and about 90% of those with pain have chronic pain conditions (3–6 months or more) [6]. Combined with deteriorating functional capacity, untreated pain leads to further reduction of cognitive function and may trigger neuropsychiatric symptoms [7]. Having pain may increase the risk of institutionalization in nursing homes, hospitalizations, both leading to prescription of PIMs (e.g., antipsychotics) [4,8]. As people with advanced dementia are often unable to describe their symptoms, they are also unable to convey the intensity, quality, location, and duration of pain [9]. In addition, they may not be able to report pain relieving effects or adverse effects of pain treatment especially when strong opioids are combined with other CNS-active drugs [10,11]. Adverse drug reactions in older adults have been found to be prevalent (4 hospitalizations /1000 person-years) [12], serious (among top 10 common causes of death) [13] and expensive (annual costs between $30 billion to $180 billion) [14,15].

Multimorbidity is defined as the co-occurrence of two or more CCC including dementia [3]. Only 5% of people with dementia are not affected by other chronic illness [3]. Although heterogeneous, CCC is often related to advanced age and treatment must be individually tailored using a patient-centred approach emphasizing individual needs adopted within integrated, coordinated care pathways [16,17]. Since trials often exclude people with co-occurring illness and those older than 65 years, clinical trial evidence for those living with dementia is severely lacking [5,18]. This means that the clinical evidence for recommended treatments, including pain management plans, are often based on trials where older adults with CCC are excluded, resulting in potentially poor generalizability to the actual population. The International Consortium for Health Outcomes Measurement proposed outcomes relevant to older adults including polypharmacy, mood and emotional health, pain, and time spent in hospital [19].

Dementia reduces the total life expectancy, but patients do not necessarily die from the disease, they die with it [20]. The life expectancy after diagnosis is on average 4.5 years, but can extend to 11 years, depending on patients’ age at the time of diagnosis and the presence of CCC, such as hypertension or diabetes. Dementia has distressing emotional, social, and economic consequences for the person affected, their informal caregivers (e.g., family, friends) and the healthcare systems which support them [21]. Most people with dementia are cared for at home by informal caregivers living with or near them and often experience high caregiver burden [22,23]. Circumstances account for a triple burden: the patient suffers more than necessary by dementia and CCC; informal caregivers suffer the excessive burden of home care, and the increasing number of people with dementia makes current treatment provision models unsustainable. For instance, a growing number of people with dementia are admitted to hospitals and receive mechanical ventilation over time on intensive care units without substantial improvement in survival [24].

These conditions are disabling and costly and the presence of pain increases the likelihood of developing CCC leading to polypharmacy with PIMs [25]. The single elements of this complex puzzle are much debated, and only few existing integrated healthcare system databases can be used as a research platform to examine the concurrent effects of these factors leading to better pain treatment plans [26]. Many military Veterans are at high risk for persistent pain due to age, injury or medical illness [27,28]. Concerns about inadequate treatment of pain—accompanied by evidence about the analgesic efficacy of opioids—has led to an increase in the use of opioid medications to treat chronic pain in the United States (US) Veterans Health Administration (VHA) and other healthcare systems [29]. Furthermore, dementia prevalence is similar in Veteran and civilian populations although, the risk of dementia is likely increased by traumatic brain injury [30]. Given the large integrated VHA system, and the availability of an integrated electronic health record, the VHA provides an unique opportunity to investigate the prevalence and associations between pain, CCC, dementia and PIMs.

Based on the importance to older adults in the International Consortium for Health Outcomes Measurement report, our specific aim is to investigate the relationship between use of pain medications and centrally acting PIMs with diagnosed dementia, level of pain, and CCC 90-days prior to hospitalization. We hypothesize that Veterans ≥65 years of age with a history of moderate to severe musculoskeletal pain, admitted to VHA facilities, are prescribed PIMs (defined using the American Geriatrics Society (AGS) Beers Criteria^®^) which differs by dementia status and pain intensity after controlling for CCC and socio-demographic factors.

## 2. Materials and Methods

### 2.1. Participants

This study used data extracted from the VHA national corporate data warehouse (CDW) for the 1,684,266 veterans who had one or more musculoskeletal diagnoses and at least one pain intensity rating of four or higher, indicating moderate to severe pain intensity, during outpatient visits between October 2012–30 September 2013 [31,32]. From this cohort, this study then selected all veterans who were admitted to VHA inpatient acute care units (*N* = 254,893).

As the Beers Criteria^®^ is for older adults, the cohort was restricted to veterans 65 years of age or older at the first admission, with complete data on key demographics (*N* = 117,715). Data on socio-demographic characteristics were obtained from the CDW. Co-occurring medical and mental health conditions (CCC) were identified from one year before inclusion in the cohort to the date of first admission. Prescriptions were extracted from VHA pharmacy for a 90-day time window prior to first admission. Pain intensity ratings were extracted from the VHA Vital Signs file. Patient-reported pain intensity on a 0 (no pain) to 10 (worst pain imaginable) numeric rating scale at each clinical encounter when vital signs are taken. Consistent with VHA policy guidance, pain intensity ratings of 4 or greater are considered to be an indication of clinically actionable pain, and ratings of 4 to 6 are classified as moderate pain intensity, while ratings of 7 or greater are considered as severe pain [33]. Accordingly, we further excluded 21,491 veterans with no or mild pain prior to first hospitalization (score <4 or unavailable), resulted a final analytic sample of 96,224, representing 81.7% of the eligible older veterans.

### 2.2. Study Procedure

Dementia, hereafter referred to as Alzheimer’s Disease and Related Dementia (ADRD) was defined based on the having one inpatient or one outpatient ICD-9 code listed in Appendix A between January 2000 to September 29, 2013 before admission.

Socio-demographic characteristics and comorbidity included age (in 5 year increments from 65 years old and then ≥90 years old), sex, race/ethnicity (Non-Hispanic Whites; Non-Hispanic Blacks; Hispanic, and other or decline to respond), and marital status (currently married versus not). To address the indication for which an analgesic or CNS-active medication may be used, we extracted relevant musculoskeletal diagnoses, medical, mental health and substance use disorder conditions listed in Table 1 (ICD-9 codes listed Appendix A). A flowchart of the sampling process is presented in Figure 2. The Charlson Comorbidity Index (CCI) was used as a measure of comorbidity.

### 2.3. Ethics

The Musculo-Skeletal Diagnoses Cohort has been approved by the Institutional Review Boards of the Veterans Administration Connecticut Healthcare System (01869 CB0020) and the Yale School of Medicine (1407014321) and has been granted a HIPAA waiver and waiver of informed consent.

### 2.4. Statistical Analyses

Baseline characteristics of the cohort by pain intensity were presented as frequency (percentages) and means (±standard deviations, SD) or median (Intra-Quartile Range, IQR) and compared with Student t-test for continuous variables and Chi-square test for categorical variables. Prevalence of pain medications and CNS PIMs within 90-days of hospitalization by pain intensity (moderate versus severe pain intensity) were compared with chi-square tests.

Logistic regression was used to estimate the odds of receiving of each medication within 90-days of hospitalization; there was a separate model for each medication. Adjusted Odds Ratio (aORs) and 95% CIs were derived for pain intensity, ADRD and their interaction, after adjusting for the socio-demographic characteristics and CCI. As a measure of polypharmacy, we estimated the aOR of receipt of three or more classes of medications for the same predictors and covariates.

All the statistical analyses were conducted using SAS software version 9.4 (SAS Institute, Cary NC, USA, 2012). The hypotheses were tested at a two-sided significance level of α = 0.05.

## 3. Results

The majority of this community-dwelling cohort was male (97.5%) with a mean age of 74 years (8 SD) with 26.6% of subjects aged 85 years or older; 70% reporting severe, versus moderate, pain (Table 1). Veterans were generally diagnosed with multiple pain conditions, with particularly high rates for three clusters of pain conditions, namely back pain, nontraumatic joint pain and osteoarthritis. Veterans with severe, versus moderate, pain intensity were significantly more likely to be diagnosed with each of these painful conditions, although Veterans with severe pain were no more likely to have any diagnosed musculoskeletal disorders than those with moderate pain. Further, those with greater pain intensity were significantly more likely have a diagnosis of the medical and mental health conditions examined, including ADRD. The exceptions were stroke or cerebrovascular disease. Overall disease burden measured by the Charlson Comorbidity Index was significantly greater among those reporting severe versus moderate pain intensity (Table 1).

Prescription of pain medications significantly differed by pain intensity, with opioids and NSAIDs being lower for those reporting severe pain, while acetaminophen and gabapentinoids were more commonly prescribed for those with severe pain intensity. Other CNS-active PIMs were also prescribed significantly more frequently for those reporting severe pain (Table 2). Similarly, 27.7% of veterans were exposed to polypharmacy based on the classes of pain and CNS-active medications studied, with a significantly higher proportion of those reporting severe pain intensity.

When adjusted in a full multivariable model (Table 3), veterans with ADRD have a significantly lower aOR (0.8; 95% CI 0.87 to 0.94) for prescription of opioids and NSAIDs (aOR 0.93; 95% CI 0.89 to 0.97), while reporting severe pain intensity was associated with significantly higher aOR of prescription of acetaminophen and gabapentinoids. For CNS-acting medications both having ADRD and reporting severe pain intensity had significantly higher aORs of receiving a prescription for all the classes of analgesics and other PIMs examined (Table 3). For polypharmacy, based on the receipt of three or more classes of medications, the aOR were significant for having ADRD (aOR 1.58; 95% CI 1.52 to 1.64) and for reporting severe pain (aOR 1.94; 95% CI 1.88 to 2.01).

When the interaction between ADRD status and pain intensity was added to each of the medication models, for the analgesic medications only gabapentinoids had a significant interaction (*p* = 0.044) (Table 4), such that for those living with ADRD and reporting severe pain there was a higher aOR (1.71; 95% CI 1.57 to 1.86) relative to those with ADRD reporting moderate pain; while Veterans without a ADRD diagnosis reporting severe pain had an aOR of 1.55 (95% CI 1.49 to 1.61) relative to non-ADRD diagnosed veterans reporting moderate pain. For all other pain medications, the increased odds of receipt for those with severe pain did not differ by ADRD status. However, for the CNS-active medications both antipsychotics and antidepressants had significant interactions (both *p* < 0.001) (Table 4).

For prescription of antipsychotics, among those with an ADRD diagnosis the pain intensity reported was not associated (aOR 0.97; 95% CI 0.87 to 1.08), while for Veterans without an ADRD diagnosis reporting severe pain had a significantly increased aOR (1.34; 95% CI 1.24 to 1.44) relative to non-ADRD diagnosed veterans reporting moderate pain. On the other hand, for prescription of antidepressants both those with and without ADRD were at increased aOR of receipt but the magnitude of the aOR was greater those Veterans without ADRD (Table 4).

For polypharmacy, based on the receipt of three or more classes of medications, the interaction was not significant (*p* = 0.86) resulting in similar aORs for veterans with diagnoses ADRD reporting severe pain (aOR 1.93; 95% CI 1.79 to 2.09) and for veterans without an ADRD diagnosis reporting severe pain (aOR 1.95; 95% CI 1.87 to 2.02).

## 4. Discussion

Earlier studies investigated the prescriptions of CNS active drugs in veterans and found that 77% who were treated with opioids also received psychotropic medication, such as antidepressants and anxiolytics, often leading to polypharmacy with potential side-effects [29].In this study, we, investigate the relationship between pain medications and levels of pain intensity in veterans with ADRD and CCC, as we hypothesize that those with dementia and pain may differ in pain management and CNS-acting PIMs. In this cohort of older veterans with musculoskeletal diagnoses reporting moderate or severe intensity pain, 70% reported severe pain intensity. In our sample, veterans diagnosed with ADRD reported greater pain intensity compared to those without ADRD. One in three with ADRD and severe pain intensity had an increased likelihood for CNS-acting medications, including opioids. Findings are of key importance for the clinicians because the PIMs and polypharmacy often worsen the daily life for people with dementia and reduce their quality of life and dignity. Although it is understandable that providers may prescribe analgesics and other CNS-acting medications in efforts to manage pain, our data encourage reconsideration of this approach when there is a lack of evidence of effectiveness and risk of harms. Proper assessment and re-assessment of pain considering CCC are prerequisites for proper treatment that reduces reliance on risky medications and encourages less risky interventions, especially non-pharmacological approaches.

### 4.1. Veterans and Pain

The high prevalence of pain among veterans in care in the VHA is well documented, with estimates suggesting that as many as 50% of male veterans, and as many as 77% of female veterans experience persistent pain [34]. Recently, a US Centers for Disease Control and Prevention Morbidity and Mortality Weekly Report further highlighted that veterans are at heightened risk for chronic pain [35]. Adding to the challenges of managing pain among veterans are high rates of co-occurring mental health and substance use disorders and co-prescribing of medications to manage these conditions [29]. As early as 1998, the VHA recognized high rates and complexity of pain and chronic pain among veterans and gaps in timely and equitable access to high quality pain care and established the VHA National Pain Management Strategy, a comprehensive approach for addressing the pain care needs of veterans. Particularly innovative at the time was introduction of a Stepped Care Model for Pain Management that highlighted the important role of primary care providers and teams to assess and manage most common pain conditions in that setting by employing low intensity and low risk approaches [36]. As already noted, despite this important initiative and evidence that implementation of the model is associated with reduced opioid prescribing, and the publication of clinical practice guidelines to encourage judicious use of opioid therapy and other risky medications [37], and institution of a comprehensive Opioid Safety Initiative [38], rates of prescribing of opioid therapy and benzodiazepine co-prescribing continued to escalate until recently [39]. VHA’s continued efforts to educate both providers and patients about potential risks of harms associated with long-term and high dose opioid therapy for the management of pain, as well as efforts to closely monitor and address polypharmacy, particularly co-prescribing of benzodiazepines and other central nervous system depressants are models for other health systems to emulate.

### 4.2. Dementia and Pain

Dementia is not a condition commonly associated with pain [40]. However, people with dementia often have concomitant joint pain, osteoarthritis and neuropathy, which combined increase the likelihood for chronic pain. We previously demonstrated that people with severe dementia and mixed dementia (Alzheimer’s dementia and vascular dementia) are at high risk to suffer from severe pain especially, related to musculoskeletal system [41]. To assess pain in people who are unable to give valid self-report, a considerable number of pain assessment tools were developed and tested during the last decades [8]. Most instruments follow the recommendations of the AGS panel and base their pain evaluation on the presence of or change in facial expression, defence and vocalization, observed during activities of daily living [42]. The Mobilization-Observation-Behavior-Intensity-Dementia Pain Scale (MOBID-2) is an instrument validated in people with dementia and responsive to pain treatment [43]. Using the MOBID-2 Pain Scale, Ersek et al., included US nursing home patients with ARDR and found that the residents’ usual pain intensity was mild (mean = 1.6/10) and intermitted (70%), but some (45%) experienced moderate to severe pain intensity. In another trial of 327 nursing home patients with dementia and behavioral disturbances, 19.3% were prescribed opioids, and of these, 79.4% were still in pain [44]. Although important, pain studies in nursing homes cannot directly be compared to studies conducted in home-dwelling people with ADRD, but comparable data are scarce [5]. In this context, several open questions need attention. For instance, pain is a challenging concept of physical, social, spiritual and psychological components (total pain) [45]. Results of pain studies are usually challenging to compare because most are lacking the crucial information whether pain intensity was assessed before or after pain treatment was initiated.

### 4.3. Complex Chronic Conditions (CCC)

In addition to dementia, veterans in our study have a Charlson Comorbidity Index of 3.9 (standard deviation ± 3.0), with a significant difference between those with moderate and severe pain (<0.0001). The Charlson Comorbidity Index is meant for the long-term prognosis of lethality for comorbid patients and is based on a point scoring system (from 0 to 40) for the presence of specific diseases. For its calculation, the points are accumulated for specific diseases, as well as the addition of a single point for each 10 years of age for patients of ages above forty years. The estimate of mortality at 3–4 points is approximately 52%. 90% of the veterans in our study have hypertension, 50% coronary artery diseases, 45% chronic obstructive pulmonary, and 49% diabetes leading to high risk for vascular dementia associated with chronic pain [41]. Patients with CCC have the highest care needs which should be individually tailored [46]. Rather than a fragmented, disease-specific approach, a patient-centered approach emphasizing individual needs must be adopted within integrated, coordinated care pathways [16]. As demonstrated in this study, most veterans with dementia have multimorbidity, and many have highly complex care needs. Since trials often exclude people with co-occurring illness, clinical evidence in dementia is severely lacking. This means that the clinical evidence for recommended treatments is often based on trials where people with co-occurring illness are excluded, resulting in potentially poor generalizability to the actual population. Future areas of research should include deprescribing trials for opioids and CNS-acting PIMs.

### 4.4. Dementia, Opioids and Pain Management

Prior to hospital admission, almost 40% of this veteran cohort were treated with opioids and co-analgesics such as gabapentinoids (22%). For those reporting severe versus moderate pain, there were increased likelihoods to be treated with any analgesic. Prescription rates for acetaminophen and NSAIDs are high and actual use is likely higher since our approach did not permit consideration of medications prescribed and filled outside the VHA or over-the-counter use. Until recently, use of opioid analgesics has rapidly increased in this population [47,48], but efficacy and safety concerns must be considered [49]. In Norway, the use of buprenorphine for older adults and people with dementia has increased, in part due to the easily administered transdermal patch formulation changed weekly. However, our double-blinded trial of buprenorphine in people with advanced dementia found increased risk of adverse events and the adverse symptoms that overlapped with common behavioral disturbances in dementia, such as changes in personality, confusion, sedation, or somnolence [10,11]. This suggests that people with dementia may experience unexpected adverse symptoms that may go unnoticed and result in hospital admission [50,51].

In general, opioid prescription rates vary widely between and within countries and regions. A Polish nursing home study, including people with mild ADRD, showed that less than 3% were treated with an opioid [52]. Almost 90% with severe ADRD did not receive any analgesic treatment compared to people with no ADRD (76%) [52]. Low opioid prescription (2.6%) was also documented in a study from Italy investigating hospitalized geriatric patients with ADRD [53]. In contrast, Nordic countries found little differences in the prescribing rates of opioids between people with and without ADRD [48,54]. In Norway (2011) 24% of nursing home patients with ADRD received any opioid [48]. In Denmark (2010), the comparable number was 38% and home-dwelling people with ADRD were more likely to receive opioids compared to cognitive intact (27.5% vs. 16.9%, respectively) [54].

In the US, Mehta et al., investigated a nursing home cohort (*N* = 734,739) and found that severe dementia was associated with lower opioid prescribing [55]. However, from 2011 to 2017, there was a 7.5% relative reduction in any opioid use among people with no ADRD (from 48.2% to 44.6%); 14.1% reduction for those with mild dementia (from 38.9% to 33.4%); 13.1% reduction for moderate dementia (from 28.3% to 24.6%), and a 10.8% reduction among people with severe dementia (from 24.1% to 21.5%) [55]. While the overall use of opioids in community-dwelling older adults increased from 2006 to 2013 [56], declining trends for opioid prescribing were reported in nursing home patients with and without dementia from 2011 to 2017 [55], potentially reflecting a change in prescribing practice as a result of increased attention to the prevention of opioid addiction and opioid overdose deaths associated with the opioid epidemic in the US. Despite a shift in prescribing practice, the likelihood of receipt of opioids remains lower in people with ADRD compared to those without ADRD in nursing homes [55,56,57].

### 4.5. Dementia and Psychotropic Drug Prescription

Preventable medical errors rank behind heart disease and cancer as the third leading cause of death in the US [58]. The simultaneous use of multiple medications can lead to dangerous drug interactions, adverse outcomes, and challenges with adherence [59]. In this study, we find that almost one in three veterans with moderate to severe pain from musculoskeletal diagnoses are using three or more concomitant psychotropic drugs. This is especially true for veterans who have both severe pain intensity and ADRD. This is of concern as people with ADRD may be unable to report potential side-effects of the treatment. In older adults and people with ADRD, psychotropic drugs are often prescribed without proper symptom assessment, without any clear indications, and treatment persists for longer than recommended [60]. Almost 36% of the veterans in our study received antidepressants, sedatives (19%) and antipsychotics (6%). Both depression and serious mental illness, as well as the treatment of these conditions, are more frequently described in those with severe pain intensity. A Federal Practice article by Battar et al. describes the implementation of a Clinical Program to Improve Patient Safety by Deprescribing Potentially Inappropriate Medications and Reducing Polypharmacy [59]. The systematic assessment of polypharmacy and reduction of potentially inappropriate medications using VIONE has benefited approximately 60,000 veterans with more than 128,000 medications deprescribed, yielding more than $4 million in annualized cost avoidance [59]. Meanwhile, our controlled nursing home intervention trials demonstrated that “less is more” both in regard to systematic collegial medication review of antihypertensives and psychotropic drugs [61,62,63,64]. We emphasize the interplay of pain and CCC in light of polypharmacy as drug penetration across the blood-brain barrier (BBB) for those with ADRD differs considerably from normal aging [65]. These differences can complicate the treatment of CNS diseases such as pain, delirium and behavioral disturbances [65].

### 4.6. Evidence-Based Non-Opioid and Nonpharmacological Approaches

The US National Pain Strategy (NPS) recommends that health systems adopt integrated, evidence-based, patient-centered and multimodal approaches for management of pain and co-occurring conditions [66]. In this context, the NPS encourages efforts to reduce reliance on risky medications, particularly long-term opioid therapy and promotion of integrated models of care that emphasize use of non-opioid medications and nonpharmacological approaches, including complementary and integrative health approaches. The VHA’s Stepped Care Model of Pain Management is cited as a promising model for reducing overprescribing of medications and promotion of effective and less risky nonpharmacological approaches, and there is early evidence of the success in achieving these aims [67,68]. In the recent past, VHA has adopted a Whole Health Initiative that aims to promote patient-centered care and increased use of evidence-based nonpharmacological approaches, perhaps especially for the management of pain and co-occurring mental health conditions [69]. A VHA sponsored State-Of-The-Art conference identified a broad array of nonpharmacological approaches that were determined to have sufficient evidence of efficacy and effectiveness to support their widespread adoption in VHA facilities [70]. An important tri-US government agency research partnership, the National Institutes of Health- Department of Defense-Veteran’s Administration (NIH-DOD-VA) Pain Management Collaboratory aims to support the conduct of pragmatic clinical trials of integrated models of care and nonpharmacological approaches for the management of pain and co-occurring conditions in VHA and military treatment facilities [71,72]. This and other VHA policy and practice initiatives hold promise for reducing the gap between evidence of the effectiveness and implementation across the VHA.

### 4.7. Strengths and Limitations

This report has strengths and limitations. Firstly, it uses a large cohort of over 96,000 older community-dwelling veterans in the VHA with pharmacy and diagnoses information, as sizable proportion 85 years or older (26.6%). This veterans cohort differs than the general population especially by being primarily male (97.5%) and represents slightly higher in black Americans (16.2%). A few more limitations and qualifiers must be mentioned. For instance, our cohort includes entirely those who were hospitalized, and findings may not be generalizable to the larger proportion of veterans in care in VHA with painful musculoskeletal conditions and ADRD. We also do not know the diagnoses upon admission; some may have been ADRD and/or pain-related (e.g., long bone fracture in a frail elderly veteran), and we do not know how medications were adjusted during the hospitalization. We should also acknowledge that we could not capture medications that were prescribed and filled outside VHA, delivered over-the-counter, use of alcohol and other illicit substances. Moreover, this cohort reported moderate to severe pain intensity making veterans at higher risk for some medications, despite implementation of a comprehensive pain management plan across the VHA that encourages judicious use of medications and promotes less risky approaches [66,67]. The design of observing medications prior to the first hospitalization and including only those with a hospitalization, we selected older veterans who may have had more CCC. However, this ensured that the medications and pain intensity ratings were prior to hospitalization and not the results of hospitalization. This observational study cannot infer causation, rather provide associations of real-world prescribing practices.

## 5. Conclusions

In our sample of older veterans hospitalized in the VHA with previously diagnosed musculoskeletal disorders reporting moderate or severe pain, those with diagnosed ADRD reported significantly greater pain intensity compared to people without ADRD. Further, ADRD and severe pain intensity had an increased likelihood for CNS-acting drug use, including opioids, leading to an increase in the PIM burden. Future trials on the causation between ADRD and CNS-acting drugs should be undertaken before implementation of comprehensive guidelines. Regular assessment and re-assessment of pain among older persons with CCC, patient-centered tapering or discontinuation of opioids, alternatives to CNS-acting PIMs and use of non-pharmacological approaches should be considered.

## Figures and Tables

**Figure 1 brainsci-11-00086-f001:**
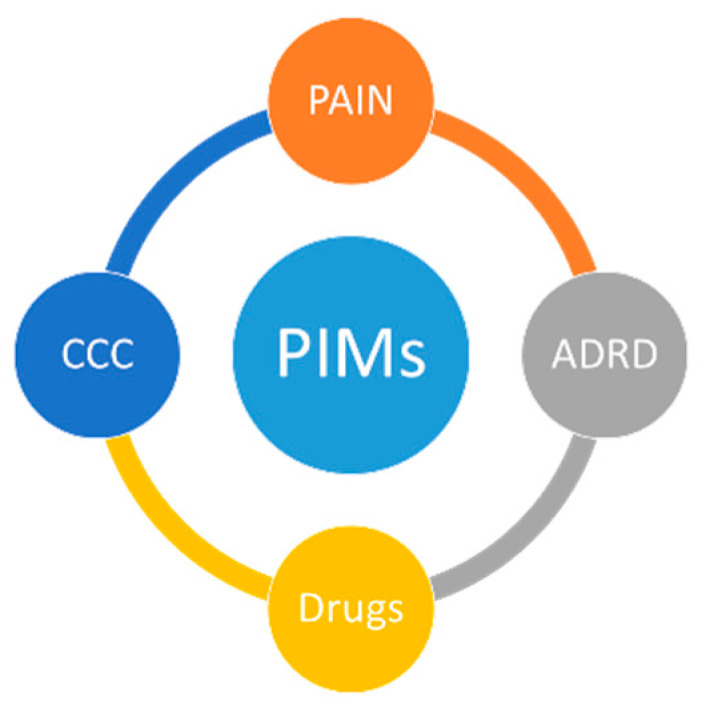
Relationship between dementia, pain, chronic complex conditions (CCC), polypharmacy and potential inap-propriate medication (PIM).

**Figure 2 brainsci-11-00086-f002:**
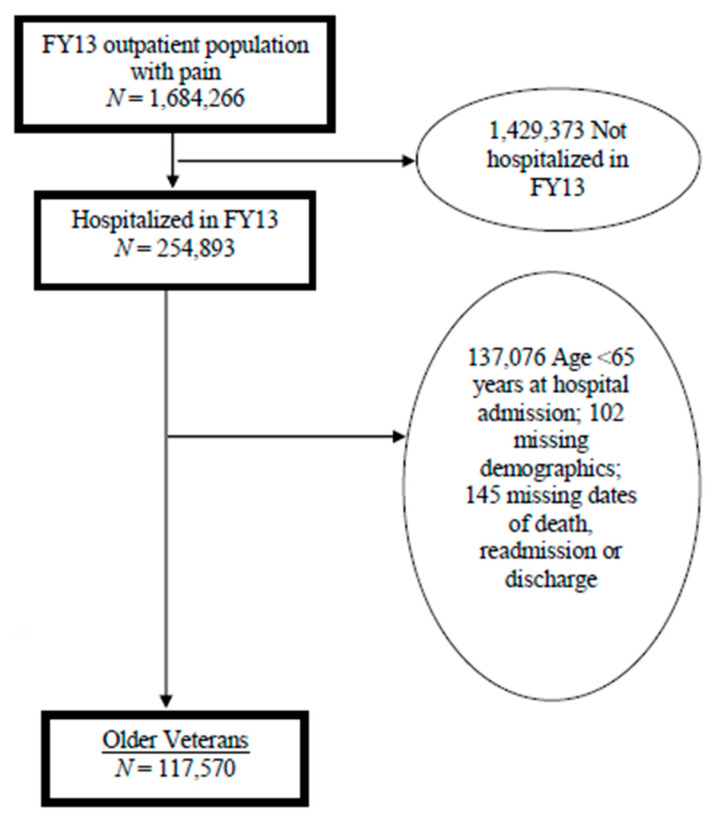
A flowchart for analytic cohort.

**Table 1 brainsci-11-00086-t001:** Characteristics of US veterans with chronic pain prior to hospital admission during fiscal year 2013: in overall sample and according to pain intensity * (*N* = 96,224).

Characteristics by Domain	Overall *N* = 96,224	Moderate Pain *N* = 28,810	Severe Pain *N* = 67,414	*p* Value ^†^
Demographic				
Age (yr), Mean ± SD	74.0 ± 8.0	73.8 ± 8.0	74.0 ± 8.0	<0.0001
*N* (%), 65–69	39,516 (41.1)	11,261 (39.1)	28,255 (41.9)	<0.0001
70–79	31,154 (32.4)	9466 (32.9)	21,688 (32.2)	
85–89	21,412 (22.3)	6703 (23.3)	14,709 (21.8)	
90+ yr	4142 (4.3)	1380 (4.8)	2762 (4.1)	
Female	2390 (2.5)	613 (2.1)	1777 (2.6)	<0.0001
Race/ethnicity				
Non-Hispanic White	73,312 (76.2)	23,226 (80.6)	50,086 (74.3)	<0.0001
Non-Hispanic Black	15,600 (16.2)	3570 (12.4)	12,030 (17.8)	
Hispanic	4859 (5.1)	1283 (4.5)	3576 (5.3)	
Other/declined to report	2453 (2.6)	598 (2.5)	1491 (2.6)	
Currently married	48,903 (50.8)	15,249 (52.9)	33,654 (49.9)	<0.0001
Musculoskeletal pain				
Back pain	55,485 (57.7)	14,353 (49.8)	41,132 (61.0)	<0.0001
Nontraumatic joint damage	80,666 (83.8)	23,031(79.9)	57,635 (85.5)	<0.0001
Osteoarthritis	57,219 (59.5)	15,799 (54.8)	41,420 (61.4)	<0.0001
Any sites of MSD pain	73,410 (76.3)	21,887 (76.0)	51,523 (76.4)	0.1264
Mental health conditions				
Post-traumatic Stress Disorders	19,871 (20.7)	4811 (16.7)	15,060 (22.3)	<0.0001
Major depressive disorders	16,135 (16.8)	3921 (13.6)	12,214 (18.1)	<0.0001
Serious mental illnesses	8777 (9.1)	2105 (7.3)	6672 (9.9)	<0.0001
Alcohol use disorders	16,343 (17.0)	4045 (14.0)	12,298 (18.2)	<0.0001
Substance use disorders	7223 (7.5)	1336 (4.6)	5887 (8.7)	<0.0001
Other clinical conditions				
Alzheimer’s disease and related dementias	17,079 (17.8)	4870 (16.9)	12,209 (18.1)	<0.0001
Chronic liver disease	6639 (6.9)	1665 (5.8)	4974 (7.4)	<0.0001
Chronic Obstructive Pulmonary Disorder and Allied Conditions	43,257 (45.0)	12,080 (41.9)	31,177 (46.3)	<0.0001
Chronic renal failure	26,285 (27.3)	7351 (25.5)	18,934 (28.1)	<0.0001
Congestive heart failure	24,530 (25.5)	6670 (23.2)	17,860 (26.5)	<0.0001
Coronary artery diseases	48,394 (50.3)	14,091 (48.9)	34,303 (50.9)	<0.0001
Delirium	9089 (9.5)	2242 (7.8)	6847 (10.2)	<0.0001
Diabetes	47,245 (49.1)	13,316 (46.2)	33,929 (50.3)	<0.0001
Hypertension	86,334 (89.7)	25,465 (88.4)	60,869 (90.3)	<0.0001
Parkinson diseases	2823 (2.9)	746 (2.6)	2077 (3.1)	<0.0001
Stroke or Cerebrovascular Disease	466 (0.48)	136 (0.47)	330 (0.49)	0.7209
Traumatic brain injury	733 (0.76)	173 (0.60)	560 (0.83)	0.0002
Charlson comorbidity index	3.9 ± 3.0	3.7 ± 2.9	3.9 ± 3.0	<0.0001

Abbreviations: MSD, Musculoskeletal disorders; Values represent mean ± SD or Frequency (%); * Moderate: PIR 4–6; Severe: PIR 7–10; ^†^ Based on Student t-test for continuous variables and Chi-square test for categorical variables, except otherwise indicated.

**Table 2 brainsci-11-00086-t002:** Prevalence of pain medications among older US veterans with chronic pain prior to hospitalization during fiscal year 2013, N (%).

Medication	Overall 96,224 (100.0)	Moderate Pain *N* = 28,810 (29.9)	Severe Pain *N* = 67,414 (70.1)	*p* Value ^†^
Opioids	36,623 (38.1)	30,636 (38.7)	5987 (35.1)	<0.001
Gabapentinoids	20,961 (21.8)	16,824 (21.3)	4137 (24.2)	<0.001
Acetaminophen	27,056 (28.1)	21,954 (27.7)	5102 (29.9)	<0.001
NSAIDs	18,625 (19.4)	15,835 (20.0)	2790 (16.3)	<0.001
Subclasses of CNS drugs				
Antipsychotics	5715 (5.9)	3701 (4.7)	2014 (11.8)	<0.001
Antidepressants	34,273 (35.6)	8944 (31.0)	25,329 (37.6)	<0.001
Sedatives/hypnotics	18,404 (19.1)	4677 (16.2)	13,727 (20.4)	<0.001
Classes ≥ 3 (range: 0–7)	26,632 (27.7)	5525 (19.2)	21,107 (31.1)	<0.001

^†^ Derived from Chi-square-test (DF = 1) on proportions of each class by pain intensity.

**Table 3 brainsci-11-00086-t003:** Association between ADRD and pain intensity for pain medication and central nervous system acting medications use during 90-days prior to hospitalization in FY 2013 among 96,224 Veterans 65 and older with moderate to severe pain.

**Pain Medication Use, aOR (95% CI)**				
**Predictor**	**Opioids**	**Gabapentinoids**	**Acetaminophen**	**NSAIDs**
(1) ADRD diagnosis vs. non-ADRD	0.90 (0.87, 0.94)	1.30 (1.25, 1.36)	1.08 (1.04, 1.12)	0.93 (0.89,0.97)
(2) Severe pain vs. Moderate pain	1.93 (1.87,1.99)	1.58 (1.52, 1.64)	1.62 (1.57,1.67)	1.36 (1.31, 1.41)
**Central Nervous System Acting Medications, aOR (95% CI)**				
	**Antipsychotics**	**Antidepressants**	**Sedative/Hypnotics**	
(1) ADRD diagnosis vs. non-ADRD	3.44 (3.25, 3.65)	2.36 (2.28, 2.45)	1.38 (1.32, 1.44)	
(2) Severe pain vs. Moderate pain	1.20 (1.13, 1.28)	1.34 (1.30, 1.38)	1.35 (1.30, 1.40)	

Abbreviations: aOR, Adjusted Odds Ratio; CI, Confidence Intervals; ADRD Alzheimer’s Disease and related dementias. aOR was estimated using a logistic regression of each drug prescription during the 90-days prior to hospitalization as a binary outcome adjusted for 5-yr age groups, sex, marital status, race/ethnicity groups, and Charlson Comorbidity Index.

**Table 4 brainsci-11-00086-t004:** Interactions associations between ADRD by pain intensity for pain medication and central nervous system acting potentially inappropriate medications use during 90-days prior to hospitalization in FY 2013 among 96,224 veterans 65 and older with moderate to severe pain.

**Pain Medication Use, aOR (95% CI)**					
		**Opioids**	**Gabapentinoids**	**Acetaminophen**	**NSAIDs**
*p* value for ADRD by Pain level interaction		0.134	0.044	0.457	0.830
ADRD	Severe vs. Moderate pain	2.04 (1.89,2.20)	1.71 (1.57, 1.86)	1.66 (1.54,1.80)	1.37 (1.24, 1.51)
Non-ADRD	Severe vs. Moderate pain	1.91 (1.85, 1.98)	1.55 (1.49, 1.61)	1.61 (1.55, 1.67)	1.36 (1.30, 1.41)
**Central nervous System Acting Medications, aOR (95% CI)**					
		**Antipsychotics**	**Antidepressants**	**Sedative/Hypnotics**	
*p* value for ADRD by Pain level interaction		<0.001	<0.001	0.503	
ADRD	Severe vs. Moderate pain	0.97 (0.87, 1.08)	1.18 (1.10, 1.26)	1.38 (1.27,1.51)	
Non-ADRD	Severe vs. Moderate pain	1.34 (1.24,1.44)	1.39 (1.34, 1.43)	1.34 (1.29,1.40)	

Abbreviations: aOR, Adjusted Odds Ratio; CI, Confidence Intervals; ADRD Alzheimer’s Disease and related dementias. aOR was estimated using a logistic regression of each drug prescription during the 90-days prior to hospitalization as a binary outcome adjusted for 5-yr age groups, sex, marital status, race/ethnicity groups, and Charlson Comorbidity Index.

## Data Availability

All Department of Veterans Affairs (VA) research is intramural, meaning only VA employees can conduct research under VA’s sponsorship. Non-VA researchers can access VA data through collaborative research studies or by becoming a VA employee. VA encourages collaboration with academic institutions and other agencies when a VA investigator has a substantive role in con-ducting the research. The data are not publicly available due to containing HIPPA information that could compromise research participant privacy/consent.

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
