# Peer review of "Pain, Complex Chronic Conditions and Potential Inappropriate Medication in People with Dementia. Lessons Learnt for Pain Treatment Plans Utilizing Data from the Veteran Health Administration"

_brainsci, 2021, doi:10.3390/brainsci11010086_

Round 1

Reviewer 1 Report

The Authors presented a review describing polypragmasy and politherapy in people, particularly Veterans, diagnosed with Alzheimer's disease and other related dementias.

This paper seems to be interesting, however some issuse need to be raised

  1. The Authors provided information that among the study group also women were found. However, analyzing the results, there is no information whether there were any significant differences between men and women given with various drugs, especially pain-relievers. It would give the possibility to look at the problem more broadly
  2. Pain is a subjective phenomenon and to rely solely on a numerical scale of pain intensity given by the patient may be misleading and may consequently result in inappropriate therapy. Therefore, did the Authors verify such data? if yes, what method was used?
  3. Most of the people in this study, if not every, use analgesics and other centrally acting drugs for a quite long time period. Is it known for the Authors whether these drugs were taken before or after the diagnosis of Alzheimer or ARDR?

Author Response

We thank the Reviewer for their comments.  As noted to another reviewer the English has been revised throughout by Prof Allore, a native English speaker. We address their 3 main points below, some of which were revised in response to the other reviewers.

#1 Unfortunately as seen in Table 1, only 2.5% of the cohort were women.  Section 4.7 lists a limitation “This Veterans cohort differs than the general population especially by being primarily male (97.5%)”.

As we would be underpowered to analyze by sex, and this would not be a pre-planned comparison, we feel that it is best to note this as a limitation.  However, as we only included older Veterans described in Section 2.1, this does represent the proportion of Veterans that are women.

#2 As described in Section 2.1 “Pain intensity ratings were extracted from the VHA Vital Signs file. Patient-reported pain intensity on a 0 (no pain) to 10 (worst pain imaginable) numeric rating scale at each clinical encounter when vital signs are taken.”  This is the VHA approved pain rating scale used for clinical care.  As noted in the Discussion, we go into detail about the subjective nature of pain reporting.

#3 We do not know how long medications were used.  In Section 2.1 we write “Prescriptions were extracted from VHA pharmacy for a 90-day time window prior to first admission.” All cohort members this study were admitted to VHA inpatient acute care units and we did not include medications during of after the hospitalization.  In Section 2.2 we write “Dementia, hereafter referred to as Alzheimer’s Disease and Related Dementia (ADRD) was defined based on the having 1 inpatient or 1 outpatient ICD-9 code listed in Supplementary Table 1 between January 2000 to September 29, 2013 before admission.”  Thus, given nearly 13 years in which to receive a ADRD ICD-9 code, it is highly likely the majority of Veterans met the ADRD criteria well before the 90-day medication exposure window.  Please see Section 4.7 in which we wrote “The design of observing medications prior to the first hospitalization and including only those with a hospitalization, we selected older Veterans who may have had more CCC.  However, this ensured that the medications and pain intensity ratings were prior to hospitalization and not the results of hospitalization.  This observational study cannot infer causation, rather provide associations of real-world prescribing practices.”

Reviewer 2 Report

The manuscript titled as: Pain, complex chronic conditions and potential inappropriate medication in people with dementia. Lessons learnt for pain treatment plans utilizing data from the Veteran Health Administration by Husebo et al., aims to investigate the relationship between receipt of pain medications and centrally (CNS) acting PIMs among Veterans diagnosed with dementia, pain intensity, and CCC 90-days prior to hospitalization. Overall, the authors have rationalized the study well, and described their results adequately; and it can be of interest to readers and researchers which operate in the biomedical field. However, I have several comments:

  1. Manuscript needs a thorough check of English grammar and font in terms of a journal style.
  2. The data is well presented, but sometimes it is confusing because of too long phrases.

Author Response

Thank you for the positive comments.  We have had Dr. Heather Allore, a co-author and a native English speaker do a through revision of the English including shortening longer sentences.

Reviewer 3 Report

The work reports a “specific aim is to investigate the relationship between use of pain medications and centrally acting PIMs with diagnosed dementia, level of pain, and CCC 90-days prior to hospitalization.” This is a relevant point in managing old people, and this paper discuss the theme appropriately and exhaustively.

There are only few of minor concerns

Page 6 “Prescription of pain medications significantly differed by pain intensity, with opioids and NSAIDs being lower for those reporting severe pain, while acetaminophen and gabapentinoids were more commonly prescribed for those with severe pain intensity”. These data are also repeated in table 2. This sentence could be more explained, because it could be interpreted as if those who have the most pain have not received the most potent analgesics. Could you please better explain the meaning of the sentence?

Page 7. “When the interaction between ADRD status and pain intensity was added to each of the medication models, for the analgesic medications only gabapentinoids had a significant interaction (p=0.044) (Table 4)”.  Could you comment on this aspect?

Page 9   I strongly agree with Authors on the sentence “As demonstrated in this study, most Veterans with dementia have multimorbidity, and many have highly complex care needs. Since trials often exclude people with co-occurring ill-ness, clinical evidence in dementia is severely lacking. This means that the clinical evidence for recommended treatments is often based on trials where people with co-occur-ring illness are excluded, resulting in potentially poor generalizability to the actual population.” So, considering that there are large differences in opioid prescription across the world, namely in EU countries, taking into account also the Metha’s study (page 10)  which in the US finds an inverse prevalence between ADRD and opioid use, what do you think about the role of opioids in pain management in a population with CCC and ADRD?

Finally, I agree with the Authors: “This observational study cannot infer causation, rather provide associations of real-world prescribing practices.”, and therefore I suggest for the future, to study causation between ADRD and a number of possible drugs, acting on CNS, before debate the implementation of comprehensive guide-lines. Underlining this aspect could improve the readability of the article

Author Response

Comments and Suggestions for Authors

The work reports a “specific aim is to investigate the relationship between use of pain medications and centrally acting PIMs with diagnosed dementia, level of pain, and CCC 90-days prior to hospitalization.” This is a relevant point in managing old people, and this paper discuss the theme appropriately and exhaustively.

Response: We thank the reviewer for their suggestions each of which are addressed below.

There are only few of minor concerns

Page 6 “Prescription of pain medications significantly differed by pain intensity, with opioids and NSAIDs being lower for those reporting severe pain, while acetaminophen and gabapentinoids were more commonly prescribed for those with severe pain intensity”. These data are also repeated in table 2. This sentence could be more explained, because it could be interpreted as if those who have the most pain have not received the most potent analgesics. Could you please better explain the meaning of the sentence?

Response: Table 2 is the prevalence of the medications by pain severity.  The next paragraph is for the multivariable model results in Table 3, where it shows that severe pain has a higher adjusted odds of all medications. 

Page 7. “When the interaction between ADRD status and pain intensity was added to each of the medication models, for the analgesic medications only gabapentinoids had a significant interaction (p=0.044) (Table 4)”.  Could you comment on this aspect?

Response: Thank you, page 7 now reads: When the interaction between ADRD status and pain intensity was added to each of the medication models, for the analgesic medications only gabapentinoids had a significant interaction (p=0.044) (Table 4), such that for those living with ADRD and reporting severe pain there was a higher aOR (1.71; 95% CI 1.57 to 1.86) relative to those with ADRD reporting moderate pain; while Veterans without a ADRD diagnosis reporting severe pain had an aOR of 1.55 (95% CI 1.49 to 1.61) relative to non-ADRD diagnosed Veterans reporting moderate pain. For all other pain medications the increased odds of receipt for those with severe pain did not differ by ADRD status.

Page 9   I strongly agree with Authors on the sentence “As demonstrated in this study, most Veterans with dementia have multimorbidity, and many have highly complex care needs. Since trials often exclude people with co-occurring ill-ness, clinical evidence in dementia is severely lacking. This means that the clinical evidence for recommended treatments is often based on trials where people with co-occur-ring illness are excluded, resulting in potentially poor generalizability to the actual population.” So, considering that there are large differences in opioid prescription across the world, namely in EU countries, taking into account also the Metha’s study (page 10)  which in the US finds an inverse prevalence between ADRD and opioid use, what do you think about the role of opioids in pain management in a population with CCC and ADRD?

Response: This is very complex because people with ADRD (and CCC) do not tolerate the opioid treatment and side effects overlap with dementia related behavioral disturbances. In addition, they are not judged to have better pain relief. The first author B. Husebø has an in press article in BMC Medicine on a placebo-controlled cross-over acetaminophen study that demonstrates that pain intensity is the same in treatment/placebo groups. However, we do not yet have a citation. Thus, we suggest the need deprescribing studies for people with dementia and opioids because older adults with dementia are historically excluded from such trials.  We have revised the end of section 4.3 to: “As demonstrated in this study, most Veterans with dementia have multimorbidity, and many have highly complex care needs. Since trials often exclude people with co-occurring illness, clinical evidence in dementia is severely lacking. This means that the clinical evidence for recommended treatments is often based on trials where people with co-occurring illness are excluded, resulting in potentially poor generalizability to the actual population. Future areas of research should include deprescribing trials for opioids and CNS-acting PIMs.”

 Finally, I agree with the Authors: “This observational study cannot infer causation, rather provide associations of real-world prescribing practices.”, and therefore I suggest for the future, to study causation between ADRD and a number of possible drugs, acting on CNS, before debate the implementation of comprehensive guide-lines. Underlining this aspect could improve the readability of the article

Response: we have revised the Conclusion to “Future trials on the causation between ADRD and CNS-acting drugs should be undertaken before implementation of comprehensive guidelines. Regular assessment and re-assessment of pain among older persons with CCC, patient-centered tapering or discontinuation of opioids, alternatives to CNS-acting PIMs and use of non-pharmacological approaches should be considered.”